# Deep learning models for forecasting dengue fever based on climate data in Vietnam

Van-Hau Nguyen[1], Tran Thi Tuyet-Hanh[2]*, James Mulhall[3], Hoang Van Minh[2], Trung Q. Duong[3]*, Nguyen Van Chien[4], Nguyen Thi Trang Nhung[2], Vu Hoang Lan[2], Hoang Ba Minh[5], Do Cuong[6], Nguyen Ngoc Bich[2], Nguyen Huu Quyen[7], Tran Nu Quy Linh[8], Nguyen Thi Tho[9], Ngu Duy Nghia[9], Le Van Quoc Anh[10], Diep T. M. Phan[11], Nguyen Quoc Viet Hung[8], Mai Thai Son[3]

1 Hungyen University of Technology and Education, Hungyen, Vietnam, 2 Hanoi University of Public Health, Hanoi, Vietnam, 3 Queen's University Belfast, Belfast, United Kingdom, 4 Hanoi University of Science and Technology, Hanoi, Vietnam, 5 Bac Ha International University, Hanoi, Vietnam, 6 University of Connecticut, Storrs, Connecticut, United States of America, 7 Vietnam Institute of Meteorology, Hydrology and Climate Change, Hanoi, Vietnam, 8 Griffith University, Brisbane, Queensland, Australia, 9 National Institute of Hygiene and Epidemiology, Hanoi, Vietnam, 10 Ho Chi Minh City University of Transportation, Ho Chi Minh, Vietnam, 11 University of Science, Ho Chi Minh, Vietnam

* tth2@huph.edu.vn (TTTH); trung.q.duong@qub.ac.uk (TQD)

**Data Availability Statement:** All relevant data are within the manuscript and its Supporting Information files.

## Abstract

### Background

Dengue fever (DF) represents a significant health burden in Vietnam, which is forecast to worsen under climate change. The development of an early-warning system for DF has been selected as a prioritised health adaptation measure to climate change in Vietnam.

### Objective

This study aimed to develop an accurate DF prediction model in Vietnam using a wide range of meteorological factors as inputs to inform public health responses for outbreak prevention in the context of future climate change.

### Methods

Convolutional neural network (CNN), Transformer, long short-term memory (LSTM), and attention-enhanced LSTM (LSTM-ATT) models were compared with traditional machine learning models on weather-based DF forecasting. Models were developed using lagged DF incidence and meteorological variables (measures of temperature, humidity, rainfall, evaporation, and sunshine hours) as inputs for 20 provinces throughout Vietnam. Data from 1997–2013 were used to train models, which were then evaluated using data from 2014–2016 by Root Mean Square Error (RMSE) and Mean Absolute Error (MAE).

### Results and discussion

LSTM-ATT displayed the highest performance, scoring average places of 1.60 for RMSE-based ranking and 1.95 for MAE-based ranking. Notably, it was able to forecast DF incidence better than LSTM in 13 or 14 out of 20 provinces for MAE or RMSE, respectively.

**Funding:** TTTH, HVM, TQD, STM received funding from the Newton Fund Vietnam: Research Environment Links, through British Council for the period of 2020-2021. The Grant ID 528154944, titled "Building Public Health Resilience in Vietnam: An Early Warning System using Artificial Intelligence" was funded by the UK Department for Business, Energy and Industrial Strategy and delivered by the British Council. For further information visit the Newton Fund website (www. newtonfund.ac.uk) and follow via Twitter: @NewtonFund The funder had no role in study design, data collection and analysis, decision to publish, or preparation of the manuscript. No authors received a salary from the funder.

**Competing interests:** The authors have declared that no competing interests exist.

Moreover, LSTM-ATT was able to accurately predict DF incidence and outbreak months up to 3 months ahead, though performance dropped slightly compared to short-term forecasts. To the best of our knowledge, this is the first time deep learning methods have been employed for the prediction of both long- and short-term DF incidence and outbreaks in Vietnam using unique, rich meteorological features.

## Conclusion

This study demonstrates the usefulness of deep learning models for meteorological factor-based DF forecasting. LSTM-ATT should be further explored for mitigation strategies against DF and other climate-sensitive diseases in the coming years.

### Author summary

Dengue fever (DF) represents a significant health burden worldwide and in Vietnam, which is forecast to worsen under climate change. The development of an early-warning system for DF has been selected as a prioritised health adaptation measure to climate change in Vietnam. This study aimed to use deep learning models to develop a prediction model of DF rates in Vietnam using a wide range of climate factors as input variables to inform public health responses for outbreak prevention in the context of future climate change. The study found that LSTM-ATT outperformed competing models, scoring average places of 1.60 for RMSE-based ranking and 1.90 for MAE-based ranking. Notably, it was able to forecast DF incidence better than LSTM in 12 or 14 out of 20 provinces for MAE or RMSE, respectively. Moreover, LSTM-ATT was able to accurately predict DF incidence and outbreaks up to 3 months ahead, though performance dropped slightly compared to short-term forecasts. This is the first time deep learning methods have been employed for the prediction of both long- and short-term DF incidence and outbreaks in Vietnam using unique, rich climate features, and it demonstrates the usefulness of deep learning models for climate-based DF forecasting.

## 1. Introduction

Dengue fever (DF) is a climate-sensitive, vector-borne disease caused by the dengue virus, transmitted primarily by *Aedes aegypti* and *Aedes albopictus* mosquitoes [1]. *Ae. aegypti* are particularly suited to urban environments, where there is an abundance of human hosts, few predators, and a wide range of potential breeding sites such as drains, tires, and water containers [2]. Symptoms of DF include flu-like symptoms such as fever, headache, joint pain, nausea, and vomiting. Severe DF (dengue haemorrhagic fever) can be fatal and may present with plasma leakage, respiratory distress, organ damage, and internal bleeding [3]. Vietnam experienced an average of 80,938 reported confirmed DF cases annually during the period from 1997–2016, representing a significant impact on public health. The burden of DF is forecast to worsen throughout the country, and temperatures in the whole country (especially southern and central regions) are predicted to become significantly more suited to DF transmission due to climate change [4]. Therefore, an effective early-warning system for DF will help to inform public health responses for outbreak prevention and has been identified as one of the prioritized health adaptation measures to climate change in Vietnam [4].

Previous studies have attempted to elucidate the relationships between meteorological factors (i.e., weather factors) and DF incidence in Vietnam and other affected countries [5–13]. Such research is useful in designing effective DF forecasting models. Multiple studies have found a positive correlation between precipitation and DF, with a lag-time of 0–3 months between high rainfall and rise in case numbers [5–9]. However, others found no significant correlation [10,11] or a negative association for a 2-month lag-time [12]. In the studies examined, minimum temperature was consistently reported as positively correlated with DF incidence for 1–2 month lags [8,10,12,13]. Average monthly or weekly temperature was reported as positively correlated at 0–2.5 month lags [5–7,9] or not significantly associated [11]. Temperature and rainfall analyses received the most coverage, however other analyses involved humidity, evaporation, sunshine hours, wind speed, and El Niño events. Relative humidity was reported as positively associated with DF in the same month by some studies [7,9,12] and negatively correlated by others [11]. When relative or minimum humidity was lagged by 1–3 months, it was only reported as positively correlated [8,12]. Sunshine hours were reported as both correlated [11] and inversely correlated [7] with DF incidence. Wind speed was found to be inversely correlated with DF cases for the same month [12]. Positive associations with DF were also found for same-month average evaporation [11] and El Niño events [10]. The regular findings of significant associations between meteorological factors and DF suggests that they may be useful predictors in forecasting DF incidence. However, the differences in findings also indicate that these relations may be location-specific.

A diverse range of forecast techniques has been applied to the prediction of DF from weather data both in Vietnam and internationally, such as those used in Kuala Lumpur, Malaysia [14]; Guangzhou, China [12]; Guadeloupe, France [15]; and Thailand [16]. These techniques include, but are not limited to, Poisson regression models [17,18], hierarchical Bayesian models [19], autoregressive integrated moving average (ARIMA) and seasonal ARIMA models [15,20,21], support vector regression (SVR) [22,23], least absolute shrinkage and selection operator (LASSO) regression [22,24], artificial neural networks (ANNs) [24], back-propagation neural networks (BPNNs), gradient boosting machine (GBM) [23], generalized additive models (GAMs) [16,23], and long short-term memory (LSTM) models [14,23]. The models listed all included temperature and rainfall as variables; other variables included humidity [8,14], air pressure and water pressure [23], wind speed [14], altitude, urban land cover [19], enhanced vegetation index [14], and data from nearby regions in the form of population flow [23] or spatial autoregression of DF risk [19].

In this study, we focused on deep learning models due, in part, to their advantages over traditional approaches. There are several limitations which traditional machine learning (ensemble and statistical) models face. Firstly, missing data can considerably decrease the performance of the models. Secondly, traditional models are not always able to discern complex patterns in the data. Thirdly, they are not able to work well in long-term forecasting applications. Finally, feature engineering in traditional models is carried out manually. In contrast, deep learning models can overcome the obstacles of traditional models through learning features directly from the data and learning much more complex data patterns in a more specific way [25,26].

Similar to the situation in various countries worldwide, there are no early-warning systems in place for the prediction of DF in Vietnam. This was identified as one of the prioritized adaptation measures of Vietnam in the "Climate change response action plan of the health sector in the 2019–2030, vision to 2050" [4]. Thus, the development of a DF early-warning system has the potential to be significantly impactful in reducing national morbidity and mortality. There are some existing studies which built DF prediction models in various provinces in Vietnam in the past [8,21,27]. However, these have mainly focused on the Mekong delta area in the

southern region of Vietnam. These prediction models have either been single-variate based on DF data or multi-variate based on common meteorological factors: temperature, humidity, and rainfall. More recently, Colón-González et al. [28] developed a superensemble of Bayesian generalised linear mixed models for DF forecasting up to six months in advance. The model was evaluated on all 63 provinces in Vietnam, using weather and land cover variables as predictors. To the best of our knowledge, there have been no DF forecasting models developed in Vietnam using advanced deep learning techniques such as LSTM. LSTM shows promising predictive accuracy when compared to other machine learning techniques in DF forecasting elsewhere [14,23] as well as in many other real-world problems [25,29–31]. This study aimed to develop an accurate prediction model for DF in Vietnam, using a wide range of weather factors as input variables.

**Contributions**. In this paper, advanced deep learning methods—CNN, Transformer, LSTM, and attention mechanism-enhanced LSTM (LSTM-ATT) models—were trained and evaluated on DF rates and 12 different meteorological variables (measures of temperature, humidity, rainfall, evaporation, and sunshine hours) from 1997 to 2016 in 20 of Vietnam's 63 provinces. Given the varying response of dengue incidence to meteorological factors observed in the literature across different locations and for different time lags, we trained the models for each province separately. To the best of our knowledge, this paper is the first to employ deep learning techniques to predict both long-term (three months ahead) and short-term (one month ahead) DF incidence and epidemic months in Vietnam. We evaluated our methods on a large number of provinces in Vietnam—20 different provinces spanning across three different regions with different geographical and climate conditions. From this evaluation, LSTM-ATT was found to outperform competing models and accurately forecast DF incidence throughout Vietnam.

## 2. Materials and methods

### 2.1 Study design and study site

This was a retrospective ecological study conducted in Vietnam. Vietnam is located in Southeast Asia, with a high level of exposure to climate-related hazards and extreme weather events. The Global Climate Risk Index 2020 ranked Vietnam as the sixth country in the world most affected by climate variability and extreme weather events over the period of 1999–2018 [32]. Vietnam has three main regions, Northern, Central and Southern Vietnam, which have distinctive geographical, meteorological, historical, and cultural qualities. Each region consists of subregions with further cultural and climate differences. Northern Vietnam has a humid subtropical climate with a full four seasons and much cooler temperatures than the South, which has a tropical savanna climate. Winters in the North can get quite cold, sometimes with frost and even snowfall. Snow can even be found to an extent up in the mountains of the extreme Northern regions, such as in Sapa and Lang Son province in recent years. Southern Vietnam is usually much hotter and has only two main seasons: a dry season and a rainy season. Climate change is projected to increase temperatures throughout the country as well as the severity and frequency of extreme weather events, which in turn would increase the number of people at risk of climate-sensitive diseases such as DF [4]. Under Representative Concentration Pathway 4.5, more frequent severe typhoons and droughts, longer monsoon seasons, and a sea-level rise of 55 cm are projected by the end of the 21st century. Temperatures are forecast to rise by approximately 2.2°C in northern regions and 1.8°C in southern regions, and annual rainfall by 5–15 mm. These changes in climate conditions are projected to significantly worsen the impact of DF and other communicable diseases in Vietnam [4], thus leading to the development of early-warning systems for them.

## 2.2 Data

DF is one of the prioritized climate-sensitive diseases in Vietnam. Monthly incident confirmed cases and deaths for DF in 20 provinces/cities (belonging to three main regions in Vietnam: North, Central, and South) from 1997 to 2016 were provided by the National Institute of Hygiene and Epidemiology (NIHE), which was responsible for the accuracy of the information in the database. There were 1,618,767 notified cases of DF from 1997 to 2016, with on average, about 80,938 cases per year (or 110 cases per 100,000 population). There were 1389 deaths from DF in this period with most of the deaths occurring before 2000. In 1998, the death rate of DF was especially high at 0.5 per 100,000. The incidence of DF and mortality rates increased as temperature increased and the rates in June to October were higher than in other months. Average yearly DF incidence rates were lower in northern Vietnam from 1997 to 2016, and peaked in central and southern provinces where the climate is hotter, rainier, and more humid (Fig 1). These conditions are advantageous to the spread of DF. Hence, including these meteorological factors into prediction models has the potential to improve prediction accuracy as demonstrated in previous works [8,14,23,24].

For weather data, 12 meteorological factors in the same period were collected, including measures of temperature, rainfall, humidity, evaporation, and sunshine hours (Table 1). Sunshine hours refers to the number of hours with the intensity of direct solar radiation reaching the surface equal to or greater than 0.2 calories/cm$^2$ minute. Surface is defined as 2 m above the ground. Thus, if there are thin clouds, but the solar radiation measured at the surface is greater than 0.2 calories/cm$^2$ minute, it will still be counted as sunny time. The data were provided by the Vietnam Institute of Meteorology, Hydrology and Climate Change (IMHEN).

## 2.3 Forecasting models

Since the raw datasets were in various formats, they had to be pre-processed and prepared for building prediction models (Fig 2).

**Data pre-processing.** The first step was to clean the data to ensure data integrity before building prediction models. Our datasets contained a few missing datapoints for some provinces. The missing data were imputed by using the minimum value from the same month of the last

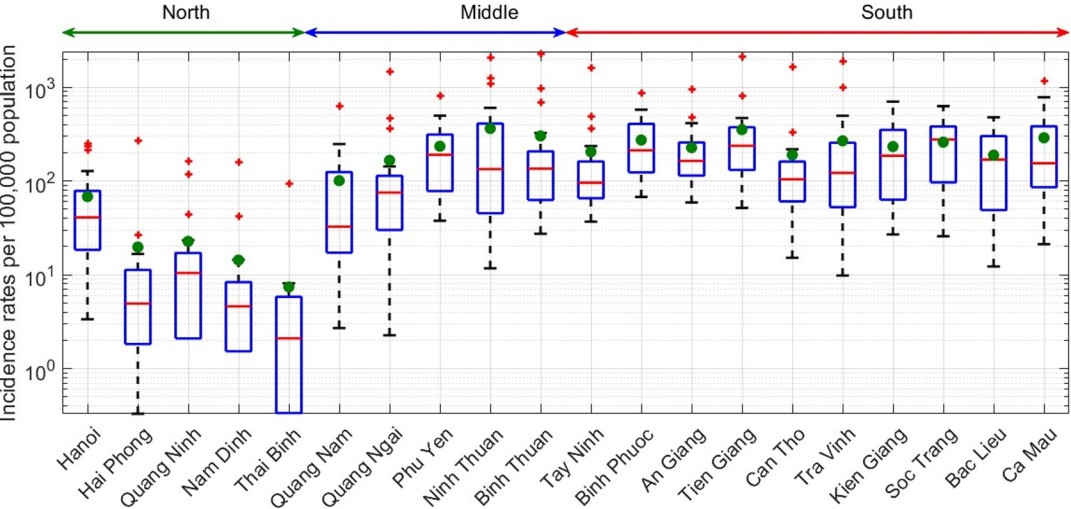

**Fig 1. Yearly DF incident cases per 100,000 population (log-scaled) for 20 different provinces in northern, central, and southern Vietnam from 1997 to 2016.** In the box and whisker plots, green dots indicate mean values.

**Table 1. Meteorological factors from the Vietnam Institute of Meteorology, Hydrology and Environment.**

| Meteorological factor | Unit | Measurement methods/detailed description of climate factors |
|---|---|---|
| Average monthly temperature | ˚C | These factors were measured in a meteorological tent at an altitude of 2m, with a frequency of four times per day. In the tent, 3 specialized thermometers were placed to measure the average temperature, the maximum temperature, and the minimum temperature. The average daily temperature value was calculated as the average of four measurements (1 am; 7 am; 1 pm; 7pm). Thus, each day had an average temperature value, a maximum temperature value, and a minimum temperature value, from which monthly data were calculated. |
| Maximum average monthly temperature | ˚C | |
| Minimum average monthly temperature | ˚C | |
| Monthly absolute maximum temperature | ˚C | |
| Monthly absolute minimum temperature | ˚C | |
| Monthly rainfall | mm | Rainfall was also measured by WMO's specialized meter and placed in a meteorological garden (close to the meteorological tent) with a frequency of measurement of four times per day. Total rainfall per day was calculated as the sum of four measurements. Thus, total monthly rainfall was calculated from the daily rainfall values. |
| Highest daily rainfall per month | mm | Selected from a series of daily rainfall in a month. |
| Number of rainy days per month | Days | Calculated from the series of daily rainfall. Number of rainy days per month is the total number of days with the rainfall greater than 0mm. |
| Monthly average relative humidity | % | Humidity was also measured in a weather tent according to WMO standards with a measurement frequency of four times per day (1 am; 7 am; 1 pm; 7 pm). The average daily relative humidity value was calculated as the average value of these four measurements. From the date data series, monthly average relative humidity was calculated. |
| Monthly minimum relative humidity | % | Daily minimum relative humidity was selected from the four measurements. From the daily data series, monthly minimum relative humidity was calculated. |
| Monthly evaporation | mm | Evaporation was also measured in a meteorological tent according to WMO standards with a measurement frequency of two times per day (7 am and 7 pm). Daily evaporation was calculated as the sum of these 2 measurements. From the daily data series, monthly evaporation was calculated. |
| Total monthly sunshine hours | Hours | Similar to the other factors, sunshine hours were also measured from a specialized meter according to WMO standards and placed in a meteorological garden to measure the total number of sunshine hours per day. From the series of daily data, the total monthly sunshine hours were calculated. |

Data for each factor was collected from 1997 to 2016. WMO = World Meteorological Organization.

two years. We found out that this scheme brings better prediction performance with our data than other common methods such as 0 and mean substitutions in preliminary experiments. Since the data contained many different features (12 weather factors and DF incidence) with different value ranges, it required normalisation. For example, total rainfall ranged from 0 mm to 3207 mm, while average temperature ranged from 3.8˚C to 31.8˚C. We normalized each data feature into a range of (0, 1) using Min-max scaling to ensure all data features were treated equally in the prediction models. Moreover, rather than predicting the numbers of DF cases each month, we predicted the incidence rate per 100,000 population to avoid the effect of population changes over time including past province expansions (e.g., the merge of Ha Noi and Ha Tay in 2008).

**Feature selection.**    For each province, we used a Random Forest Regressor from the Scikit-learn Python Library (version 0.24.2) [33] to rank the importance of all meteorological factors using Recursive Feature Elimination (RFE) and choose the top 2 features as input for prediction models. In this method, the RFE function was first trained on all meteorological factors as predictors of DF incidence by using random forest regression, then the least

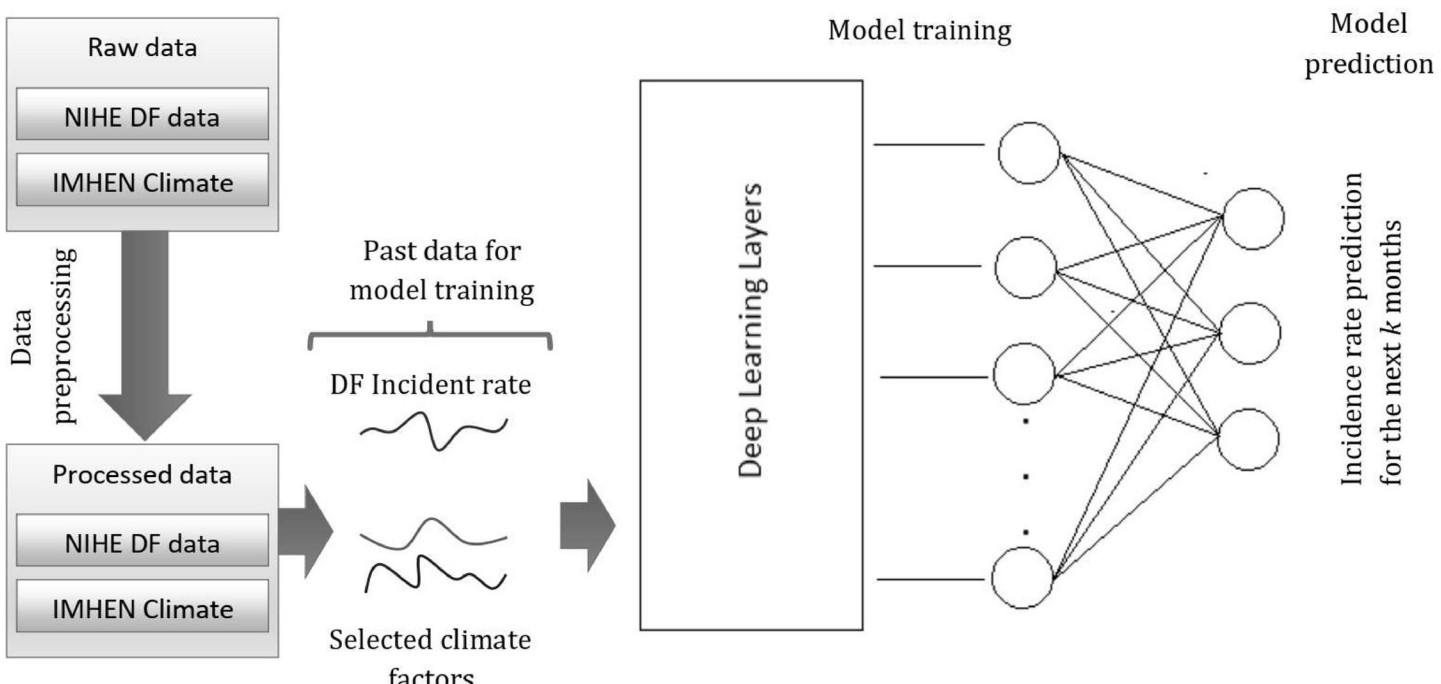

**Fig 2. Data processing pipeline.** NIHE = National Institute of Hygiene and Epidemiology. IMHEN = Vietnam Institute of Meteorology, Hydrology and Climate Change. DF = dengue fever.

important meteorological factor was removed. This process was repeated recursively until there were only the two most important features left. This helped to improve the model's efficiency and effectiveness by avoiding overfitting caused by too many input features. The full list of features for each province can be found in S1 Table.

**Performance evaluation.** Models were evaluated for predictions made one to three months (steps) in advance. Multi-step prediction refers to forecasts made more than one month in advance. We split our data into a training set (from 1997 to 2013—a total of 17 years) and a testing set (from 2014 to 2016–3 years in total) for each province. The training data were used as input to fit the parameters of the prediction models. We used RMSE and MAE as two main measures to evaluate how our forecasted incidence rates compared to the real ones in the test set for each province. In this context, MAE can be interpreted as the average absolute difference between predicted and actual DF rates over the three years test set. MAE computes the mean of the absolute errors between predicted values and corresponding real values as follows:

$MAE = \frac{1}{n}\sum_{i=1}^{n} |y_i - \hat{y}_i|$, where $y_i$ is an actual value and $\hat{y}_i$ is a predicted value.

MAE weights errors in proportion to their magnitude. RMSE, in contrast, weights larger errors more heavily than smaller errors. RMSE computes the square root of the mean of squared errors between predicted values and corresponding real values.

$RMSE = \sqrt{\frac{1}{n}\sum_{i=1}^{n} (y_i - \hat{y}_i)^2}$, where $y_i$ is an actual value and $\hat{y}_i$ is a predicted value.

Generally, lower scores in these RMSE and MAE metrics indicate a better forecasting model. As RMSE weights larger errors more than MAE, a forecast with lower RMSE and higher MAE than competing models would likely have more small-scale errors but fewer large-scale errors.

**Outbreak detection.** The ability to correctly categorise months as either outbreak (i.e., epidemic) or non-outbreak (i.e., normal) months was assessed for the LSTM-ATT model. We

set an epidemic threshold for each province by using the monthly mean and standard deviation of incidence rates for that province as in previous works [34,35]. An outbreak month is defined by an incidence rate exceeding the mean by $n$ standard deviation(s). We set $n = 1$ in our study to capture both medium and large outbreak months. Four metrics were used to assess epidemic detection, as defined below. Firstly, accuracy is defined as the ability of a model to correctly categorise future months as normal or outbreak months. Secondly, precision refers to the ratio of correctly detected outbreak months to the number of predicted outbreak months. Thirdly, sensitivity refers to proportion of outbreak months that were correctly predicted. Finally, specificity is defined as the ratio of correctly detected normal months to the total number of normal months.

$$Accuracy = \frac{Correct\ Predictions\ (Outbreak\ or\ Normal\ Month)}{Total\ Predictions}$$

$$Precision = \frac{Correct\ Predictions\ (Outbreak\ Months)}{Total\ Predicted\ Outbreak\ Months}$$

$$Sensitivity = \frac{Correct\ Predictions\ (Outbreak\ Months)}{Total\ Outbreak\ Months}$$

$$Specificity = \frac{Correct\ Predictions\ (Normal\ Months)}{Total\ Normal\ Months}$$

**Forecasting models.** ANNs [36] are a type of computational model, which imitate the information processing achieved by neurons in the human brain by making the right connections among nodes [29]. An ANN consists of three parts: a layer of input nodes, layers of hidden nodes, and a layer of output nodes. ANNs are able to successfully map nonlinear input to output by automatically extracting subtle patterns and multiple features from a large dataset through each layer. Modern ANNs have achieved state-of-the-art results in previous DF studies in different regions with different meteorological and geographic data, such as in China [22,23] and Kuala Lumpur, Malaysia [14]. Thus, in this paper, we focus on adapting these advanced prediction models to predict DF rates for Vietnam, through the use of CNNs [29], LSTM models [37] with and without attention mechanisms [30,31], and a Transformer model [30]. Additionally, a selection of traditional machine learning models—Poisson regression [38], XGBoost [39], Support Vector Regression (SVR and SVR-L) [40], and Seasonal AutoRegressive Integrated Moving Average (SARIMA) [41]—were included for comparison. Our prediction methods take DF rate and some selected weather factors as inputs and output the forecasted DF incidence rates for the next $k$ consecutive months (Fig 2). In this paper, we fixed $k = 3$ for forecasting future DF incidence up to 3 months ahead in 20 provinces. However, we also tested with $k = 6$ in Hanoi to provide an extended example.

**CNNs**: The development of CNNs was a breakthrough in ANNs, as they approached human performance in a wide range of domains including pattern recognition, natural language processing, and video processing by processing data in grid-like topology [25,29]. Thus, we adapted CNN models to cope with longitudinal data. Our CNN model consisted of 1D convolution layer, 1D max pooling layer, and one fully connected layer.

**LSTM**: Recurrent Neural Networks are another variant of ANNs specifically designed to cope with time ordered data, where nodes are connected as a directed graph along a temporal sequence [42]. In this paper, we focused on LSTM [37], one of the most successful variants of RNNs specifically designed to deal with longer dependencies in sequences [43] and reduce exploding gradients. Unlike RNNs, instead of adding regular neural units (i.e., hidden layers),

LSTM adds memory blocks. A common LSTM memory block consists of a cell state and three gates—an input gate, a forget gate, and an output gate.

**LSTM-ATT**: LSTMs can lose important information due to passing information across multiple sequence steps. To deal with this limitation, attention mechanisms were originally introduced in Machine Translation [31,33] to strengthen the power of exploiting information by generating an output at each sequence step. They have proven to be an effective approach for long input sequences. For this reason, we employed the attention technique from Luong et al. [31] to further enhance the performance of LSTM in this paper by adding an attention layer after the LSTM network, denoted as LSTM-ATT.

**Transformer**: We also considered the Transformer model [30], a recent advanced deep learning model for natural language processing, for our task. Like RNNs, the Transformer is designed to deal with sequential data. However, it does not process sequential data in order like LSTMs. Instead, the Transformer handles the sequence data by using self-attention mechanisms to learn the complex dynamics of time series data.

**Model Implementation:** We implemented the deep learning prediction models (CNN, LSTM, LSTM-ATT and Transformer) in Python 3.7.10 using PyTorch (version 1.8.1) [44] and Scikit-learn (version 0.24.2) [33] libraries. During our experiments, we tried lookback window lengths from 1 to 18 (months). We observed that the models performed best once the lookback window length was set to 3 (months). After tuning different configurations for parameters and hyperparameters, we applied the best fitting configurations as follows. For all models, the following parameters were used: *batch size* = 16, *learning rate* = 1e$^{-3}$, *dropout* = 0.1, *number of training epoch* = 300. For CNN, the following parameters were used: *number of layers* = 1, *number of each kernel* = 100, *size of each kernel is* (1, 3), (2, 3) and (3, 3). The numbers of layers and hidden sizes for LSTM, LSTM-ATT, and Transformer were optimized for different provinces and models (S2 Table).

Poisson regression, SVR, and SVR-L models were implemented in Scikit-learn (version 0.24.2) (Pedregosa et al., 2011), while XGBoost models were implemented in the XGBoost Python package (version 1.5.0) (Chen and Guestrin, 2016). For the Poisson regression models, *alpha* was set to 1e-15, and *max_iter* to 1e6. For XGBoost models, default parameters were used. For SVR, the following parameters were used: *kernel* = "rbf", *C* = 100, *gamma* = "auto", *epsilon* = 0.1. For SVR-L, the following parameters were used: *kernel* = "linear", *C* = 100, *gamma* = "auto".

SARIMA models were implemented using the SARIMAX model from the statsmodels (v0.12.2) Python library [45]. Default function parameters were used with the exception of *enforce_stationarity* and *enforce_invertibility*, which were set to false, and the models were not retrained while iterating through the test set. The order, seasonal order, and trend parameters were chosen using Bayesian model-based optimisation. This was implemented with a Tree-structured Parzen Estimator (TPE) in Optuna (version 2.8.0) [46] which aimed to find the optimum combination of parameters for each province to minimise RMSE (S3 Table). There were many parameters to optimise for the SARIMA models, which can be highly time-consuming and difficult for fine-tuning. Therefore, the decision was made to automate this process, and a TPE was chosen over grid-searching as it is less computationally expensive [46].

**Ethical consideration:** This study was approved and managed by the Hanoi University of Public Health. The study only involved analysing secondary data on DF cases and climate factors including temperature, precipitation, humidity, evaporation, and sunshine hours. No human participants were actually involved in this study. Thus, ethical approval was not required.

## 3. Results

**One step forecasting accuracy**: Overall, the deep learning models outperformed traditional models in forecasting DF incidence in all 20 provinces, as measured by RMSE (Table 2) and MAE (Table 3). Colour-coded results were used to highlight this on a province-by-province basis, instead of colour-coding across the entire range of values, as RMSE and MAE values are only directly comparable where observed incidence rates are the same. Compared to the traditional models, LSTM-ATT had lower RMSEs and MAEs in all provinces, LSTM had lower MAEs in all provinces and lower RMSEs in all but one province, CNN had lower values for both error metrics in all but three provinces, and Transformer had lower error metrics in all but four provinces.

To visualize the prediction performances of different models compared to the real incidence rates, we plotted the predicted values of the best performing models—CNN, LSTM and LSTM-ATT—as well as the actual incidence rates for all provinces during the last 36 months from January 2014 to December 2016 (S1 Fig). Plots from six different provinces were provided for an overview of the forecasting results (Fig 3), as well as complete epidemic curves for all provinces across the full 20 years of the dataset (S2 Fig). As the transformer and traditional models performed poorly, they were excluded to avoid overplotting. Overall, the prediction lines of LSTM and LSTM-ATT fit very well with the actual incidence lines for most of the provinces indicating very good prediction accuracies in these provinces. On the other hand, the performances of CNN and especially Transformer were less stable than LSTM and LSTM-ATT in most provinces.

**Table 2. Root mean square errors for all prediction models in 20 Vietnamese provinces.**

| Province | Root Mean Square Error for Each Model | | | | | | | | |
|---|---|---|---|---|---|---|---|---|---|
| | LSTM | LSTM-ATT | CNN | TF | Poisson | XGB | SVR | SVR-L | SARIMA |
| Ha Noi | 7.999 | 6.630 | 9.180 | 11.301 | 17.162 | 13.382 | 16.689 | 16.878 | 18.144 |
| Hai Phong | 0.464 | 0.529 | 0.757 | 0.748 | 0.934 | 0.657 | 6.073 | 7.938 | 2.594 |
| Quang Ninh | 1.010 | 0.961 | 1.953 | 0.930 | 1.577 | 1.277 | 3.384 | 4.072 | 1.175 |
| Nam Dinh | 0.783 | 0.797 | 0.974 | 1.008 | 0.939 | 1.156 | 1.454 | 1.578 | 0.933 |
| Thai Binh | 0.627 | 0.597 | 0.598 | 0.661 | 0.688 | 0.738 | 0.781 | 0.878 | 0.676 |
| Quang Nam | 7.382 | 6.696 | 6.890 | 12.678 | 13.504 | 11.990 | 13.969 | 15.434 | 16.448 |
| Quang Ngai | 9.288 | 8.080 | 8.874 | 8.861 | 11.113 | 9.096 | 27.721 | 37.677 | 10.181 |
| Phu Yen | 9.187 | 9.544 | 9.766 | 12.544 | 19.278 | 16.209 | 19.329 | 20.562 | 20.628 |
| Ninh Thuan | 5.064 | 3.959 | 5.140 | 8.743 | 17.260 | 24.833 | 20.274 | 12.441 | 9.027 |
| Binh Thuan | 8.364 | 8.826 | 8.259 | 12.031 | 12.949 | 10.302 | 13.880 | 14.512 | 10.120 |
| Tay Ninh | 5.123 | 3.854 | 6.538 | 6.500 | 7.350 | 9.395 | 7.213 | 9.450 | 6.600 |
| Binh Phuoc | 6.577 | 7.466 | 9.063 | 9.649 | 14.796 | 12.574 | 17.746 | 17.507 | 21.731 |
| An Giang | 5.699 | 3.907 | 3.860 | 5.461 | 9.502 | 8.672 | 7.777 | 7.954 | 10.504 |
| Tien Giang | 4.415 | 4.098 | 7.912 | 5.620 | 18.336 | 17.611 | 14.648 | 16.247 | 13.550 |
| Can Tho | 3.119 | 2.228 | 3.997 | 4.866 | 8.689 | 6.595 | 18.503 | 27.518 | 9.349 |
| Tra Vinh | 4.462 | 3.891 | 4.820 | 4.482 | 12.442 | 13.630 | 14.752 | 14.289 | 10.129 |
| Kien Giang | 2.460 | 2.976 | 4.448 | 3.892 | 16.070 | 16.809 | 16.093 | 16.455 | 5.079 |
| Soc Trang | 6.192 | 5.887 | 3.725 | 4.389 | 12.671 | 13.908 | 12.227 | 11.946 | 42.093 |
| Bac Lieu | 3.429 | 2.652 | 2.379 | 2.891 | 12.324 | 11.841 | 10.035 | 9.584 | 23.812 |
| Ca Mau | 4.490 | 4.110 | 5.499 | 9.043 | 14.720 | 20.489 | 15.279 | 15.974 | 17.736 |

Values are colour-coded for each province separately from the lowest value (darker green) to the median value (yellow) to the highest value (darker red). LSTM = long short-term memory. LSTM-ATT = attention mechanism-enhanced LSTM. TF = Transformer. CNN = convolutional neural network. Poisson = Poisson regression. XGB = Extreme Gradient Boosting. SVR = Support Vector Regressor with Radial Basis Kernel. SVR-L = Support Vector Regressor with Linear Kernel. SARIMA = Seasonal Autoregressive Integrated Moving Average.

**Table 3. Mean absolute errors for all prediction models in 20 Vietnamese provinces.**

| Province | Mean Absolute Error for Each Model | | | | | | | | |
|---|---|---|---|---|---|---|---|---|---|
| | LSTM | LSTM-ATT | CNN | TF | Poisson | XGB | SVR | SVR-L | SARIMA |
| Ha Noi | 4.926 | 3.457 | 5.065 | 5.695 | 8.397 | 7.199 | 9.077 | 9.542 | 8.637 |
| Hai Phong | 0.276 | 0.366 | 0.538 | 0.702 | 0.817 | 0.434 | 5.196 | 7.838 | 2.541 |
| Quang Ninh | 0.652 | 0.614 | 1.223 | 0.560 | 1.325 | 0.876 | 2.945 | 3.973 | 0.786 |
| Nam Dinh | 0.556 | 0.492 | 0.654 | 0.748 | 0.796 | 0.806 | 1.229 | 1.428 | 0.728 |
| Thai Binh | 0.412 | 0.432 | 0.428 | 0.468 | 0.498 | 0.420 | 0.664 | 0.803 | 0.522 |
| Quang Nam | 3.766 | 4.116 | 4.039 | 8.353 | 8.730 | 8.216 | 9.567 | 11.802 | 10.505 |
| Quang Ngai | 6.699 | 6.579 | 6.183 | 5.913 | 9.442 | 6.739 | 24.494 | 36.921 | 7.112 |
| Phu Yen | 6.604 | 7.342 | 6.433 | 10.167 | 13.429 | 11.923 | 15.608 | 17.670 | 18.062 |
| Ninh Thuan | 3.733 | 2.813 | 3.875 | 5.351 | 15.816 | 17.633 | 17.566 | 9.028 | 5.589 |
| Binh Thuan | 6.606 | 6.495 | 6.300 | 9.692 | 9.929 | 7.755 | 11.225 | 11.898 | 7.280 |
| Tay Ninh | 4.405 | 2.837 | 5.218 | 5.305 | 5.517 | 6.622 | 5.460 | 8.220 | 5.585 |
| Binh Phuoc | 5.020 | 5.353 | 6.846 | 7.546 | 10.957 | 10.042 | 14.780 | 13.715 | 16.440 |
| An Giang | 4.462 | 3.006 | 2.769 | 3.747 | 8.476 | 7.057 | 6.762 | 7.021 | 9.423 |
| Tien Giang | 3.845 | 3.371 | 6.589 | 4.876 | 15.919 | 13.528 | 10.893 | 14.204 | 10.671 |
| Can Tho | 2.611 | 1.884 | 2.911 | 4.469 | 6.725 | 4.864 | 16.782 | 27.370 | 8.148 |
| Tra Vinh | 3.143 | 2.702 | 3.528 | 4.005 | 9.376 | 9.435 | 11.766 | 11.692 | 7.984 |
| Kien Giang | 1.848 | 2.093 | 3.537 | 3.110 | 13.859 | 12.334 | 14.397 | 14.652 | 3.765 |
| Soc Trang | 4.393 | 4.540 | 3.084 | 3.304 | 10.683 | 10.326 | 10.310 | 10.283 | 36.243 |
| Bac Lieu | 2.870 | 2.160 | 2.008 | 2.207 | 11.494 | 9.399 | 9.142 | 8.897 | 19.599 |
| Ca Mau | 3.553 | 2.935 | 4.582 | 5.710 | 12.015 | 11.213 | 13.103 | 14.381 | 16.263 |

Values are colour-coded for each province separately from the lowest value (darker green) to the median value (yellow) to the highest value (darker red). LSTM = long short-term memory. LSTM-ATT = attention mechanism-enhanced LSTM. CNN = convolutional neural network. TF = Transformer. Poisson = Poisson regression. XGB = Extreme Gradient Boosting. SVR = Support Vector Regressor with Radial Basis Kernel. SVR-L = Support Vector Regressor with Linear Kernel. SARIMA = Seasonal Autoregressive Integrated Moving Average.

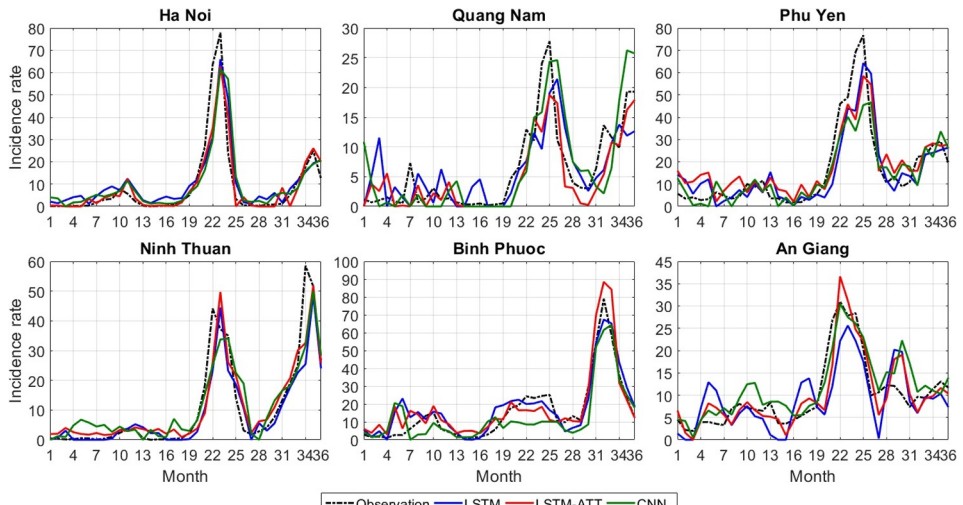

**Fig 3. Prediction performances of CNN, LSTM, and LSTM-ATT during the last 36 months in six Vietnamese provinces.** Predicted incidence rates per 100,000 population from 2014 to 2016 are shown compared to the observed incidence rates. The closer the predictions are to the observed values, the better the prediction accuracies. CNN = convolutional neural network. LSTM = long short-term memory. LSTM-ATT = attention mechanism-enhanced LSTM.

The RMSE and MAE metrics for the full set of 20 provinces in Vietnam further quantify the differences in deep learning model performance initially seen in the DF incidence plots (Fig 4). LSTM and LSTM-ATT clearly outperformed CNN and especially Transformer in most cases, indicated by low RMSE and MAE values, such as in Ha Noi and Tay Ninh. LSTM-ATT had the lowest RMSE in 10 provinces, followed by LSTM in five provinces, CNN in four provinces, and Transformer in one. For RMSE, LSTM-ATT was better than LSTM in 14 out of 20 provinces. Similarly, for MAE, LSTM-ATT had the lowest score in eight provinces, followed by LSTM in five provinces, CNN in five provinces, and Transformer in two provinces. LSTM-ATT had a lower MAE than LSTM in 13 out of 20 provinces. This shows the improvement the attention mechanism brings to the prediction accuracies of LSTM in our task.

To have a better overall view of the performance of these models on all provinces, we ranked each model from one to nine based on the RMSEs and MAEs for each province where one was the best method and nine was the worst. After that, we calculated the average ranks for all methods across all 20 different provinces. LSTM-ATT outperformed all other techniques with average rankings of 1.60 for RMSE and 1.95 for MAE (Fig 5). LSTM was the second-best method with average rankings of 2.35 and 2.20 for RMSE and MAE, respectively. The CNN model placed third, with average rankings of 3.10 and 2.70 for RMSE and MAE, respectively. The other models had worse error scores overall, with transformer ranking fourth, XGBoost fifth, Poisson regression sixth, SARIMA seventh, SVR eighth, and SVR-L nineth. Therefore, the deep learning models outperformed traditional models, and the attention mechanism improved the performance of the baseline LSTM model.

**One step outbreak prediction**: LSTM-ATT was selected for outbreak prediction due to its high performance relative to competing models. Overall, LSTM-ATT was able to predict epidemic months very well with a low incidence of false alarms (Fig 6A) and high levels of precision, accuracy, sensitivity, and specificity (Fig 6B). There was an average accuracy score (i.e., the ability to classify months as either outbreak or normal) of 0.99, and an average sensitivity score (i.e., the ability to detect outbreak months) of 0.70. However, the average sensitivity calculation is based on the five provinces where there were outbreaks, as sensitivity is undefined

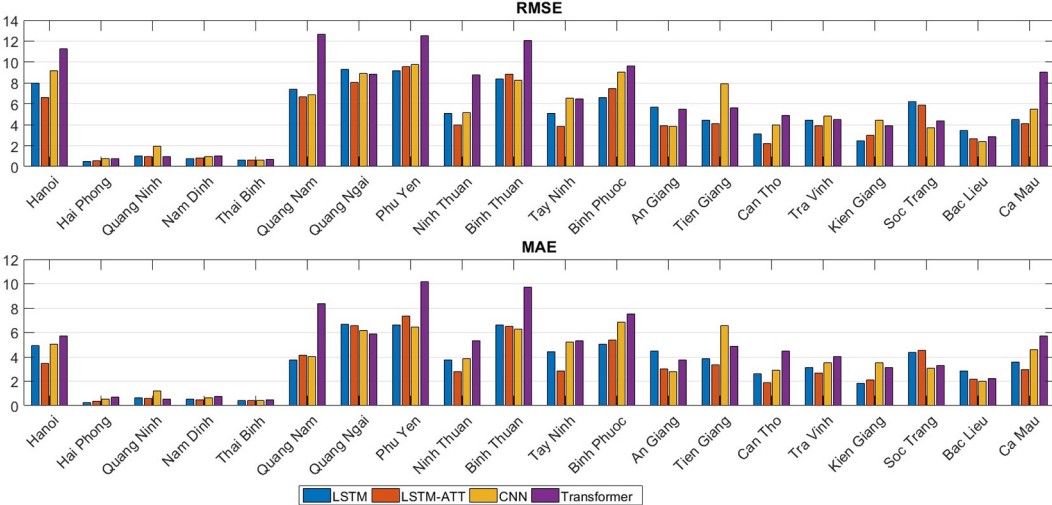

**Fig 4. RMSEs and MAEs for all models (LSTM, and LSTM-ATT, CNN, Transformer) for all 20 provinces.** The smaller the values, the better the prediction accuracies. RMSE = root mean square error. MAE = mean absolute error. CNN = convolutional neural network. LSTM = long short-term memory. LSTM-ATT = attention mechanism-enhanced LSTM.

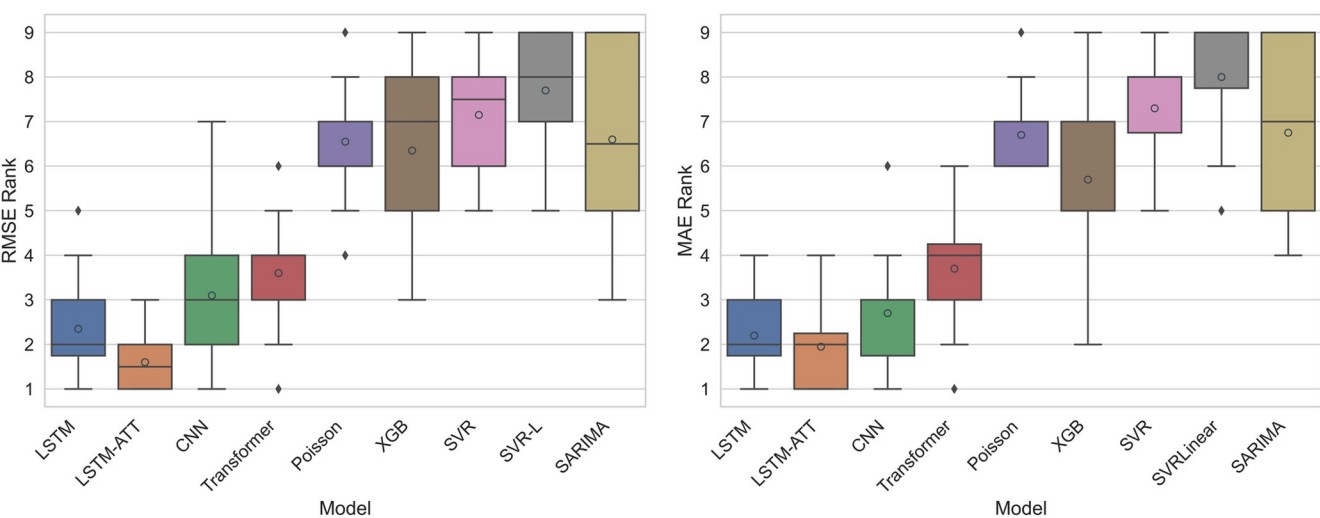

**Fig 5. DF forecasting models with RMSE- and MAE-based rankings.** Rankings are based on the relative scores for lowest RMSE or MAE in the prediction of dengue fever one month ahead. Grey-outlined circles indicate mean values. RMSE = root mean square error. MAE = mean absolute error. LSTM = long short-term memory. LSTM-ATT = attention mechanism-enhanced LSTM. CNN = convolutional neural network. Poisson = Poisson regression. XGB = XGBoost Extreme Gradient Boosting. SVR = Support Vector Regressor with Radial Basis Kernel. SVR-L = Support Vector Regressor with Linear Kernel. SARIMA = Seasonal Autoregressive Integrated Moving Average.

for all other months. Specifically, LSTM-ATT detected all true outbreak months in Ha Noi, Quang Nam, and Binh Phuoc. It missed one true outbreak month in Thai Binh and Phu Yen, and raised one false alarm in Ha Noi and Phu Yen. This meant there were precision and sensitivity scores of 0 for Thai Binh and 0.50 in Phu Yen. For all other provinces which did not have any outbreaks, LSTM-ATT was able to detect all normal months (i.e., those with no outbreaks) correctly. This led to specificity and accuracy scores of 1.0 for most provinces.

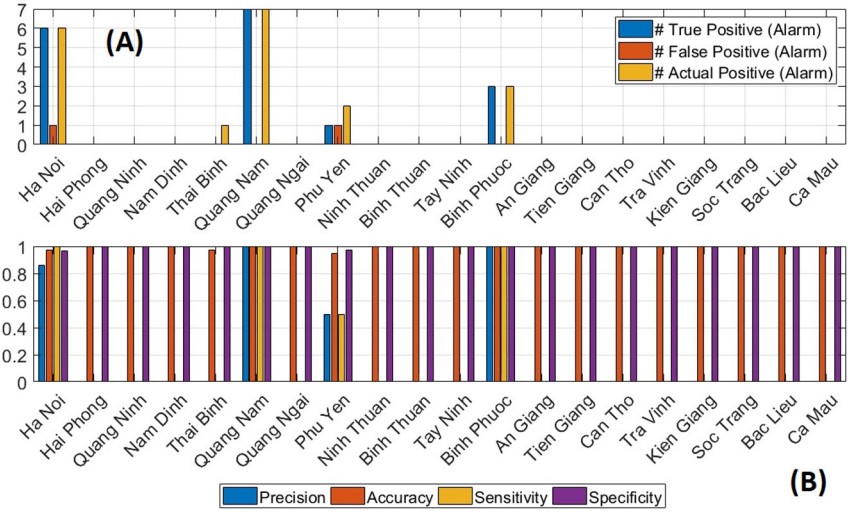

**Fig 6. Outbreak detection by LSTM-ATT.** Numbers of actual outbreak months, correct outbreak month predictions (true positive) and incorrect outbreak month predictions (false positive) for each province are shown (Fig 6A). Additionally, prediction metrics (precision, accuracy, sensitivity, and specificity) for each province are displayed (Fig 6B). If a province did not have any actual epidemic months in the evaluation period, the precision and sensitivity are not available. LSTM-ATT = attention mechanism-enhanced LSTM.

**Multi-step ahead prediction**: The performance of LSTM-ATT was then assessed for predictions 2–3 months in advance (Fig 7A). Obviously, it is harder to predict in longer term. Thus, it is unsurprising that RMSE and MAE increased for some provinces. However, for most provinces, the changes were small (or even better in a few cases) indicating very good prediction performance of LSTM-ATT. This is also observed in the plotted incidence rates. For example, in Ha Noi, Ninh Thuan, and Binh Phuoc, there were high similarities between the predicted and observed rates (Fig 7B). In most of the 20 provinces, however, there were visible reductions in performance while forecasting more months in advance (S3 Fig). Further forecasts of up to six months in advance in Hanoi showed a continuing worsening of performance (S4 Fig).

**Multi-step outbreak prediction**: As with 1-month ahead predictions, outbreak month detection was assessed for forecasts 2–3 months ahead (Fig 8). As expected, the performance dropped for some provinces when predicting further into the future (e.g., for Binh Phuoc, Quang Nam and Ha Noi). However, the overall performance was still approximately the same for almost all of the other provinces.

## 4. Discussion

This study found that LSTM-ATT frequently outperformed competing deep learning models in DF prediction and displayed a marked improvement over the basic LSTM model. Further

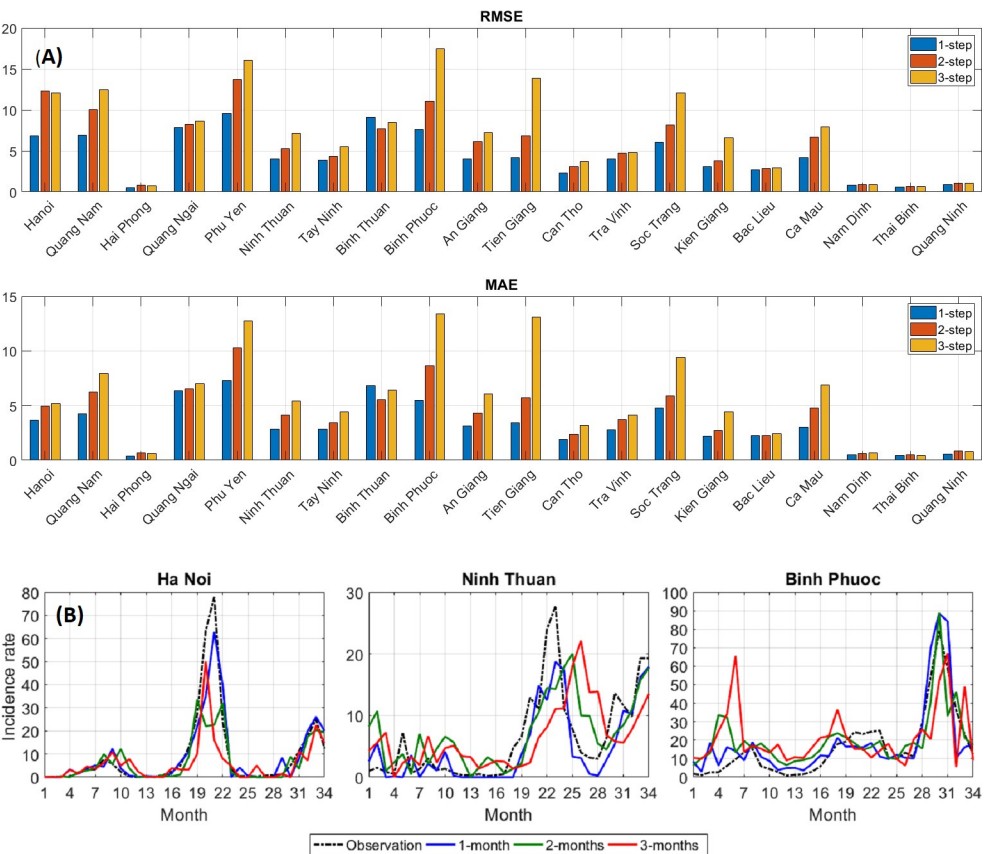

**Fig 7. Performance of multi-step ahead predictions of LSTM-ATT for all provinces.** Error metrics are displayed for all 20 provinces (Fig 7A for RMSE and middle for MAE) in addition to the predicted and observed incidence rates per 100,000 population in three provinces (Fig 7B). LSTM-ATT = attention mechanism-enhanced LSTM. RMSE = root mean square error. MAE = mean absolute error.

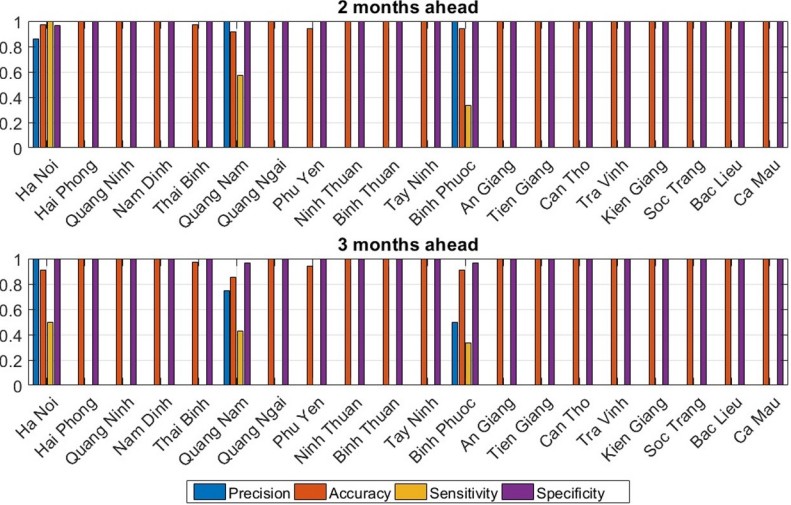

**Fig 8. Precision, accuracy, sensitivity, and specificity for multi-step ahead epidemic prediction using LSTM-ATT.** LSTM-ATT = attention mechanism-enhanced long short-term memory.

exploration revealed that LSTM-ATT could accurately forecast DF incidence and predict outbreak months up to 3 months ahead, though accuracy dropped slightly compared to short-term forecasting. While other studies have applied a country-level threshold to identify epidemic months [17], the incidence of DF in Vietnam varies across regions, provinces, and cities. Therefore, a single threshold method is not appropriate. By setting the outbreak threshold as one standard deviation above the monthly mean for a province, both medium and large-scale outbreaks were detected, which may be more useful for mitigating DF epidemics.

Meteorological factors are, in part, associated with changes in DF incidence because of their impacts on mosquito development and behaviour. The implementation of an early-warning system for DF requires it to be based on data that is widely accessible throughout Vietnam at short notice with low costs involved, and weather data and case numbers satisfy these criteria unlike other correlates of DF such as mosquito density [47]. The models in this study used a subset of rich meteorological factors including temperature, precipitation, humidity, evaporation, and sunshine hours for forecasting, as recursive feature selection identified these as the most relevant predictors out of the 12 weather variables available. Development rates increase for *Ae. aegypti* eggs, larvae, and pupae from 12˚C up to 30˚C, then drop sharply after 40˚C [48]. Additionally, biting rate may increase with temperature [49] and estimated dengue epidemic potential increases with average temperature up to 29˚C for low diurnal temperature ranges but is lower with high diurnal temperature ranges [50]. Increases in rainfall have been shown to increase mosquito density and oviposition of *Ae. aegypti*, which can facilitate endemicity [51]. The pooling of rainwater in containers and tires can create breeding grounds for mosquitoes [4], though excessively heavy rainfall, conversely, has been proposed to flush out breeding sites [12]. Furthermore, humidity is associated with increased survival of *Ae. aegypti* [52], and evaporation could impact *Aedes* mosquitoes through its effects on humidity. Previous works in Thailand [16] and Puerto Rico [53] have found models including weather data to perform worse than those that did not. The complex mechanisms described here between DF and weather could explain why deep learning models show considerable predictive ability in forecasting DF incidence—simpler models may be unable to adequately process the non-linear biological relationships. In our results, the SARIMA model only used previous DF incidence as a predictor, and performed worse than the deep learning models which

included meteorological factors. However, an evaluation of equivalent deep learning models with and without meteorological factors would be required for a true comparison.

Lookback windows from 1–18 months were tested on the deep learning models, with three months resulting in optimal performance. This corresponds well with previous correlation studies between DF and meteorological factors, which have reported time lags preceding altered DF incidence of 0–3 months for rainfall [5–9,12], 0–2.5 months for temperature [5–10,12,13], 0–3 months for humidity [7–9,11,12,27], and 0 months for evaporation [11]. Therefore, the 3-month window appears to capture the relevant delays between altered weather conditions and DF incidence, which could be due to effects on mosquito development and activity, or human behaviours such as leaving screenless windows open or spending more time outdoors.

In general, the LSTM-ATT model frequently outperformed the other deep learning models being assessed. Moreover, LSTM-ATT outperformed LSTM in 13 and 14 provinces when measured by MAE and RMSE, respectively. In Quảng Nam, the MAE was lower for the standard LSTM model, but the RMSE was lower for LSTM-ATT. As RMSE attributes greater weight to larger errors unlike the linear weighting of MAE, this suggests the LSTM-ATT model had more small-magnitude errors but fewer large-magnitude errors than the standard LSTM model. This is likely to be preferable in DF forecasting, where the underestimation of an outbreak could be catastrophic.

To the best of our knowledge, this study is the second to forecast long term DF incidence and outbreak months on a large scale in Vietnam. Disease incidence and epidemic detection remained relatively accurate for forecasts up to three months in advance, which further illustrates the utility of LSTM-ATT in DF forecasting. There are very few works exhibiting true long-term DF prediction. Colón-González et al. [28] recently developed a weather and land cover-based probabilistic superensemble of generalised linear mixed models (GLMMs) to forecast DF in all 63 provinces in Vietnam up to 6 months in advance. Average accuracy and sensitivity scores of 73% and 68% were obtained for outbreaks more than two standard deviations above the mean. As a different outbreak threshold was used and results were averaged across 1–6 months lags, direct comparisons with our results are not possible. However, the cost effectiveness analysis in the study suggests implementing the superensemble model could improve relative value in reducing the impact of DF outbreaks compared to not using a prediction model in most provinces. Therefore, future work to directly benchmark GLMM superensembles and deep learning models may be useful.

Outside of Vietnam, a few long-term weather-based DF forecasting models have been developed. Hii et al. [17] reported high prediction precision for a Poisson multivariate regression model forecasting DF outbreak months in Singapore 16 weeks in advance. The model had a Receiver Operating Characteristics (ROC) area under the curve (AUC) of 0.98 for outbreak forecasting. However, case numbers were much lower than they are in Vietnam. There was only one outbreak to assess performance on in the one-year validation period, reducing the robustness of the analysis. Shi et al. [54] employed LASSO regression to develop models for up to 3-months ahead DF forecasting in Singapore, with a MAPE of 17% for a 1-month lag and 24% for a 3-month lag. Notably, they integrated mosquito breeding index with meteorological data for predictions. Both of these studies were on a national level, while Chen et al. [55] used LASSO regression for neighbourhood level forecasting in major residential areas in Singapore. They reported AUC values of 0.88–0.76 for predictions of 1–12 weeks, respectively. Additionally, non- meteorological data was integrated in the form of cell-phone derived travel metrics, building age, and Normalised Difference Vegetation Index.

Previous studies comparing weather-based DF forecasting techniques are in agreement with our findings regarding the high accuracy of LSTM models. Xu et al. [23] found LSTM to

be superior to BPNN, GAM, SVR, and GBM techniques, with transfer learning improving predictions in lower-incidence areas. Similarly, Pham et al. [14] found a genetic algorithm enhanced LSTM model to provide better accuracy than linear regression and decision tree models. Here, we present a novel implementation of the attention-mechanism for LSTM models in the prediction of DF incidence from meteorological data, and demonstrate its improved accuracy over CNN, standard LSTM, and Transformer models. Notably, LSTM-ATT outperformed the basic LSTM model in almost all provinces, suggesting LSTM-ATT could be a more robust choice for future studies on the prediction of climate-sensitive diseases.

Surprisingly, the Transformer model performed poorly throughout the study, even though it has previously been shown to outperform LSTM-based models in some other applications [56]. In most of the cities, the Transformer performed worse, and under-fitting was observed in many of the results. The advantage of Transformer is that the model is based on self-attention. This helps the Transformer by not processing the sequential data in order and can reduce training time due to parallel computation. This advantage, however, does not appear to carry over to the research presented in this study, which might be better handled strictly in order due to the seasonality of the data. In other words, processing the input data in this paper as a whole seems ineffective.

This study had several limitations regarding alternate correlates of DF incidence, case reporting, and dengue virus serotypes. One was not accounting for various non-weather-based factors of DF transmission, such as human behaviour, travel patterns, mosquito density, dengue virus serotypes, and public health programs for DF prevention and control. These were, however, impractical to model on a national or provincial scale for an early-warning system in Vietnam. On a similar note, missing case and meteorological data may be confounding factors. Some of the differences observed between provinces may be attributable to different rates and methods of data reporting between locations. Additionally, as this was a retrospective study, all data was available in real time. Due to delays in case reporting, prospective forecasting sometimes requires predictions to be made with incomplete case data. Reich et al. [57] found this to be the case for DF forecasting in Thailand, and reported reduced model accuracy for predictions into the future as a result. Real-world implementation of the deep learning models presented in this study, therefore, may have higher errors than presented here. Lastly, multiannual spikes in DF incidence have previously been a barrier to accurate DF prediction and have been attributed to antibody-dependent enhancement following a new serotype being introduced to a region [19]. While the models presented here were only evaluated on 36 months of data, they appear to partially overcome this limitation and accurately predict large multi-annual fluctuations in cases.

## 5. Conclusion

In this study, we developed and evaluated a selection of deep learning models for the prediction of DF incidence and epidemics in Vietnam. In contrast to most existing works, which have focused on smaller study areas in Vietnam with fewer weather variables [8,21,27], our models were built upon a rich set of 12 different meteorological factors (including temperature, precipitation, humidity, evaporation and sunshine hours) and evaluated on 20 different provinces in northern, central and southern regions of Vietnam. These regions display significantly different geographical and climate conditions, allowing for a robust assessment of model performance. LSTM techniques were found to display considerable accuracy in forecasting DF incidence, with LSTM-ATT demonstrating improved prediction performance over other models in nearly all provinces. Vietnam is experiencing a digital transformation in healthcare. Digital technologies, such as AI with deep learning models for forecasting climate-

sensitive diseases, come as a promising measure to promote public health responses to climate change and enhance their efficiency. The application of LSTM-ATT in forecasting other prioritized climate-sensitive diseases in Vietnam such as influenza, diarrhoea, and malaria should be further explored.

## Supporting information

**S1 Table. Ranked features for all provinces.** Features were ranked by recursive feature elimination using a random forest regressor to rank importance. The features are listed in order from most important to least important.
(DOCX)

**S2 Table. Numbers of layers and hidden sizes for LSTM, LSTM-ATT and Transformer for all provinces.** LSTM = long short-term memory. LSTM-ATT = attention mechanism-enhanced LSTM.
(DOCX)

**S3 Table. Seasonal Autoregressive Integrated Moving Average model parameters.**
p = autoregressive term. d = differencing term. q = moving average term. t = trend (n = no trend, c = constant trend, t = linear trend, ct = constant and linear trend). P = seasonal autoregressive term. D = seasonal differencing term. Q = seasonal moving average term.
s = periodicity.
(DOCX)

**S4 Table. Root Mean Square Errors (RMSEs) for models assessed on 20 provinces in Vietnam.** North, Central, and South refer to the three major geographic regions of Vietnam.
(DOCX)

**S5 Table. Mean Absolute Errors (MAEs) for models assessed on 20 provinces in Vietnam.** North, Central, and South refer to the three major geographic regions of Vietnam.
(DOCX)

**S1 Fig. Dengue fever incidence rates in 20 Vietnamese provinces from 1997–2016.** Dengue fever rates were plotted as monthly incidence per 100,000 population.
(TIFF)

**S2 Fig. Deep learning model predictions for dengue fever rates in 20 Vietnamese provinces.** Predicted incidence rates per 100,000 population from 2014 to 2016 are shown compared to the observed incidence rates. Only the highest performing models are shown to avoid overplotting. CNN = convolutional neural network. LSTM = long short-term memory. LSTM-ATT = attention mechanism-enhanced LSTM.
(TIFF)

**S3 Fig. LSTM-ATT multi-step ahead incidence forecasting for all provinces.** Predicted incidence rates per 100,000 population from 2014 to 2016 are shown compared to the observed incidence rates. Predicted incidence is shown for forecasts made 1–3 months ahead. LSTM-ATT = attention mechanism-enhanced LSTM.
(TIFF)

**S4 Fig. Evaluation of dengue fever forecasting up to six months in advance in Hanoi.** On the left, observed dengue fever incidence is plotted as well as predictions made 2, 4, and 6 steps (months) in advance. On the right, RMSE and MAE values are shown for predictions made $k$ months in advance.
(TIF)

**S1 Data. The analyzed data as a supplementary file for easy reproducibility of the reported findings.**
(ZIP)

## Author Contributions

**Conceptualization:** Van-Hau Nguyen, James Mulhall, Hoang Van Minh, Trung Q. Duong, Nguyen Thi Trang Nhung, Mai Thai Son.

**Data curation:** Van-Hau Nguyen, James Mulhall, Hoang Van Minh, Trung Q. Duong, Nguyen Van Chien, Nguyen Thi Trang Nhung, Vu Hoang Lan, Hoang Ba Minh, Do Cuong, Nguyen Huu Quyen, Tran Nu Quy Linh, Nguyen Thi Tho, Ngu Duy Nghia, Le Van Quoc Anh, Mai Thai Son.

**Formal analysis:** Van-Hau Nguyen, Tran Thi Tuyet-Hanh, James Mulhall, Trung Q. Duong, Nguyen Van Chien, Nguyen Thi Trang Nhung, Vu Hoang Lan, Nguyen Huu Quyen, Le Van Quoc Anh, Mai Thai Son.

**Funding acquisition:** Tran Thi Tuyet-Hanh, Hoang Van Minh, Trung Q. Duong, Nguyen Ngoc Bich, Mai Thai Son.

**Investigation:** Tran Thi Tuyet-Hanh, Nguyen Ngoc Bich, Nguyen Thi Tho, Mai Thai Son.

**Methodology:** Van-Hau Nguyen, Tran Thi Tuyet-Hanh, Hoang Van Minh, Hoang Ba Minh, Do Cuong, Nguyen Ngoc Bich, Tran Nu Quy Linh, Diep T. M. Phan, Nguyen Quoc Viet Hung, Mai Thai Son.

**Project administration:** Tran Thi Tuyet-Hanh, Hoang Van Minh, Mai Thai Son.

**Resources:** Tran Thi Tuyet-Hanh, Ngu Duy Nghia.

**Software:** James Mulhall, Hoang Ba Minh, Diep T. M. Phan, Nguyen Quoc Viet Hung, Mai Thai Son.

**Supervision:** Tran Thi Tuyet-Hanh, James Mulhall, Mai Thai Son.

**Validation:** Van-Hau Nguyen, Nguyen Van Chien, Nguyen Thi Trang Nhung, Vu Hoang Lan, Hoang Ba Minh, Do Cuong, Nguyen Ngoc Bich, Nguyen Huu Quyen, Tran Nu Quy Linh, Nguyen Thi Tho, Ngu Duy Nghia, Le Van Quoc Anh, Diep T. M. Phan, Nguyen Quoc Viet Hung, Mai Thai Son.

**Visualization:** Nguyen Thi Trang Nhung, Vu Hoang Lan, Hoang Ba Minh, Do Cuong, Nguyen Huu Quyen, Tran Nu Quy Linh, Nguyen Thi Tho, Ngu Duy Nghia, Le Van Quoc Anh, Diep T. M. Phan, Nguyen Quoc Viet Hung, Mai Thai Son.

**Writing – original draft:** Van-Hau Nguyen, Tran Thi Tuyet-Hanh, James Mulhall, Hoang Van Minh, Trung Q. Duong, Nguyen Van Chien, Mai Thai Son.

**Writing – review & editing:** Van-Hau Nguyen, Tran Thi Tuyet-Hanh, James Mulhall, Hoang Van Minh, Trung Q. Duong, Nguyen Van Chien, Nguyen Thi Trang Nhung, Vu Hoang Lan, Hoang Ba Minh, Do Cuong, Nguyen Ngoc Bich, Nguyen Huu Quyen, Tran Nu Quy Linh, Nguyen Thi Tho, Ngu Duy Nghia, Le Van Quoc Anh, Diep T. M. Phan, Nguyen Quoc Viet Hung, Mai Thai Son.

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
