## [Decision Letter · Decision Letter 0]

30 Aug 2021

Dear Associate Prof. Tran Thi,

Thank you very much for submitting your manuscript "Deep learning models for forecasting dengue fever based on climate data in Vietnam" for consideration at PLOS Neglected Tropical Diseases. As with all papers reviewed by the journal, your manuscript was reviewed by members of the editorial board and by several independent reviewers. In light of the reviews (below this email), we would like to invite the resubmission of a significantly-revised version that takes into account the reviewers' comments. 

The language needs to be checked carefully.

Please upload the analyzed data as a supplementary file for easy reproducibility of the reported findings.

Please write in the cover letter the section name, page and line numbers for every change you make.

We cannot make any decision about publication until we have seen the revised manuscript and your response to the reviewers' comments. Your revised manuscript is also likely to be sent to reviewers for further evaluation.

Sincerely,

Mohamed Gomaa Kamel

Associate Editor

Samuel Scarpino

Deputy Editor

The language needs to be checked carefully.

Please upload the analyzed data as a supplementary file for easy reproducibility of the reported findings.

Please write in the cover letter the section name, page and line numbers for every change you make.

Reviewer's Responses to Questions

**Key Review Criteria Required for Acceptance?**

**Methods**

-Are the objectives of the study clearly articulated with a clear testable hypothesis stated?

-Is the study design appropriate to address the stated objectives?

-Is the population clearly described and appropriate for the hypothesis being tested?

-Is the sample size sufficient to ensure adequate power to address the hypothesis being tested?

-Were correct statistical analysis used to support conclusions?

-Are there concerns about ethical or regulatory requirements being met?

Reviewer #1: I’ll appreciate if you can explain why climate factors including temperature, precipitation, humidity, evaporation, and sunshine hours were chosen for analysis, since there were a range of climate factors, as the article cited in the introduction part. Many factors, such as wind speed and air pressure, were excluded.

The yearly data shown in Fig 1 barely have relations with the result, so it seems unnecessary.

Fig 2 can be elucidated in words.

Reviewer #2: -The study design is appropriate and clearly stated the objective.

- Comprehensive retrospective data is used and split into training and testing subsets, which was appropriate for testing the hypothesis.

Reviewer #3: The manuscript makes a good exposition of the health problem generated by dengue. Likewise, the relationship of climatic factors with dengue cases is exposed, emphasizing temperature, presipitation and others.

The objective of the study is clear when proposing a prediction model based on climatic variables. I consider it necessary to specify the study design. It is easy to understand that it is a prospective longitudinal ecological study. However, the research group needs to define its study. 

The study population is made up of data from 1997 to 2016, which is adequate. In general, the proposed statistical analysis is consistent with the objective of the study.

Reviewer #4: 1) The objective is clearly stated. However, it could be put into more relevance if the last two paragraphs of the Introduction were reorganized.

2) I would find very informative to see a figure (maybe as supplementary, or replacing Fig 1) displaying the complete epidemic curves for each province along the 20 years.

3) The data employed is very adequate. There is extensive case data all over Vietnam for a considerable span of time (20 years).

4) The tools employed are adequate and respond to the state-of-the-art in machine-learning.

5) Analyses could be improved if varying training sets were used to test how sensitive the performance and results of the models are to the training and test data.

6) k=3 looks rather short. There is no justification for this choice. It would strengthen the results showing the performance for a range of k, say 2 to 6, for different lookback window lengths.

7) What criteria were used to choose the structure and parameters of the models? For instance, lines 246-7, or 287-9. Please, explain in text.

8) An explanation about the different information captured by MAE and RMSE would help interpretating the results. Something on this regard appears in the midst of the discussion (lines 455-6), but a richer explanation should appear in Methods. Additionally, it would be informative to provide criteria on how to understand values of these statistics, besides comparing the relative values.

9) I am not sure it is worth devoting space to Figs 2-5 in the main text.

10) Results contain some description of the methods that should be moved to the method section (e.g., lines 346-56; 369-77).

11) Please, for clarity avoid using the same name for different variables. E.g., k used for the prediction window and for SDs for outbreak definition.

12) I would not call an “outbreak” each month above the threshold. I find it counterintuitive because usually an outbreak is considered as the whole “peak” that may exceed a month. Instead, referring to “epidemic months” or “outbreak months” could be more eloquent.

13) Handling of missing data: why were missing data not interpolated between adjacent months with data? This seems more appropriate than extrapolating from another year, particularly for dengue, that has such interannual variability. Please, better justify in the text or change the method. Just mentioning “We found out that this scheme brings better prediction performance” seems insufficient.

14) There are no ethical concerns.

15) Data are not made available. Given that the data used does not imply any privacy concern (because it is aggregated at a province level), I would expect that authors could meet the journal spirit of increasing data availability. Additionally, no specific web address or e-mail is given to ask for the data, as indicated in the journal guidelines.

Reviewer #5: The authors may wish to compare using deep learning models versus using simpler regression models.

Reviewer #6: The authors have clearly defined the objectives of the study and have proposed clear testable hypothesis. The study design appropriately addressed their objectives and used an appropriate population to test the hypothesis. However, additional information on the study areas, specific metrics for predictor factors ranking, and information on metrics for prediction methods should be provided.

Under 2.1 Vietnam. A brief description of main geographic/climate characteristics of the three regions studied in Vietnam is needed, this will provide readers with a better idea of the importance to include provinces representing these regions in Vietnam. It will also facilitate the interpretation of the results when different predictions are obtained in each region and climate factors are driving the differences.

This study indicates that prediction robustness was increase by analyzing 12 meteorological factors. Although, those factors were ranked per province, and the top 2 factors were selected/province to be imputed into testing models. It is not clear how the ranks were defined, were R2 coefficient of determination estimated? 

A brief description of the “outbreak prediction accuracy” method and the statistical relevance is missing, although the first result presented in Fig 1 depicts predicted incidences vs. real incidence as part of the 1-step forecasting accuracy, there is need of statistical values, e.g. 95% credible intervals, or IC values for the real incidence line depicted on the graphs.

The precision, accuracy, sensitivity, and specificity of multi-step ahead epidemic prediction methods should be described here in the methods, this information is given at the end of results when describing fig 11. It will help the reader to get this information earlier that at the result section.

Reviewer #7: The study shows for the first time the use of deep learning models for climate-based Dengue Fever forecasting. The methods are well described with details. The statistical analysis was well performed in my oppinion.

**Results**

-Does the analysis presented match the analysis plan?

-Are the results clearly and completely presented?

-Are the figures (Tables, Images) of sufficient quality for clarity?

Reviewer #1: Only the results of 6 provinces were manifested in Fig 6. Please explain what dose “the other provinces do not convey additional information”, in line 298-300, mean? It would be advantageous if a supplementary figure could be presented.

The actual outbreak of 15 provinces is zero in Fig 9. So most of precision and accuracy are zero. It makes the conclusion about the prediction ability of LSTM-ATT not so persuading.

True positives depend on rate in line 354, while false positives depend on number in line 356. It might be inconsistent.

Fig 10 shows the multi-step ahead predictions results of three provinces. It will be appreciated if the results of the other 17 provinces could be put in the supplementary figure.

Reviewer #2: yes

Reviewer #3: The results obtained in the study coincide with the proposed analysis plan. On the other hand, figure 2 is not very informative, the process could be written in the document. Figure 2 is reversed. The other images are adequate.

Reviewer #4: 16) Results for all provinces could be provided in the supplementary material.

17) What “one step” or “multi step” means should be explained in Methods (it is explained in the middle of the results (line 381), after being used paragraphs before (line 294)).

18) Most results (and figures) refer to predictions one month ahead. Initially, this is not clear at all since authors refer to “one-step” without timely explaining what a “step” is. The explanation appears paragraphs after in the midst of the results (line 381). However, in Methods (lines 240-1), it is stated that k=3 “in this paper” which, as explained there, means that predictions are made 3 months in advance. Predicting one month in advance does not look of great utility for implementing control measures. Also, generally transmission varies rather smoothly between adjacent months (Fig 6 for instance). Therefore, predicting one month in advance should not be a very difficult task, even without such complex methods. Instead, predicting 3 or, even better, 4-6 months in advance is not straight-forward and can indeed be of help for vector-control programs. It would be very informative seeing something like Fig 10(bottom) but at the very beginning of Results, for all provinces (at least in supplementary results), and including larger k’s.

19) As currently presented, figures have a poor resolution, even when downloaded.

20) x-axis labels of Fig 6 could be provided as divisor of 12 months per year; it would be more intuitive. Alternatively, the date or month of the year could be provided.

21) Fig 9: bar grouping by province results rather confusing given the blank spaces for several provinces.

22) Something similar to Fig 9 for the other models could be provided as supplementary figures to give a full account of the performance of all models.

23) y-axis should always display an axis title (e.g., Figs 7, 9, 10top, etc.).

Reviewer #5: (No Response)

Reviewer #6: Most of the results are clearly and completely presented as outlined in the study design, however additional information on some of the results may improve clarity and relevance to the current findings. 

Fig 1 depicts predicted incidences vs. real incidence as part of the 1-step forecasting accuracy, there is need of statistical values, e.g. 95% credible intervals, or IC values for the real incidence line depicted on the graphs.

The Table S1 provide a list of the top 2 geometrical factors that ranked best among 12 factors evaluated per each province. Actual rank values could be added for each selected factor for data completeness.

Reviewer #7: The figures and tables presented shows the results with quality and clarity. The authors used different type of graphics according to data. The results are clearly demonstrated.

**Conclusions**

-Are the conclusions supported by the data presented?

-Are the limitations of analysis clearly described?

-Do the authors discuss how these data can be helpful to advance our understanding of the topic under study?

-Is public health relevance addressed?

Reviewer #1: The cause of higher outbreaks in the middle part of the country isn't the purpose of the discussion. The part from line 461 to 469 links weakly with the objective of the study.

Reviewer #2: Yes, the conclusions support the data presented, but the study did not cite or compare their result with the countrywide study published in 2021, "Probabilistic seasonal dengue forecasting in Vietnam: A modeling study using superensembles."

It will be good if they can provide some insight on results comparisons from that study. 

The study results are good and can improve the paper's discussion by comparing the results with the study mentioned above.

Reviewer #3: The conclusions of the study are consistent with the objective and proposed methodology. The analyzes made it possible to establish prediction models for dengue cases.

The authors establish limitations in the study. However, I consider that it is necessary to mention that the prognostic models allow to establish the moment of occurrence of the epidemic but do not establish the place. This consideration limits the development of entomological prevention and control strategies.

The study contributes to knowledge from a new methodological point of view. However, it is similar to the considerations identified by other research. The novelty related to the study is the methodological contribution, which I consider should be published.

These procedures and results are relevant in public health as they allow the prognosis of dengue epidemics to be established.

Reviewer #4: (No Response)

Reviewer #5: (No Response)

Reviewer #6: Conclusions are supported by the data presented overall, however the limitations of analysis can be expanded to clarify their relevance on current study results and in the context of current/competing methods. Although, the contribution of this study to advance the development of deep learning methods to accurately forecast DF in Vietnam is novel and highly valuable, authors can emphasize a little more on its public health relevance regarding its potential for guiding decision-making processes.

Reviewer #7: Discussion of the results included the limitations of the methods employed and conclusions describe with clearance the findings demonstrated in the paper.

**Editorial and Data Presentation Modifications?**

Reviewer #1: Fig 10 shows the multi-step ahead predictions results of three provinces. It will be appreciated if the results of the other 17 provinces could be put in the supplementary figure.

Reviewer #2: Some figures can be transferred to supplementary files. 

Accept with Minor Revisions.

Reviewer #3: In general, the article is consistent and accurate. I recommend changing the brackets for the parentheses in line 206, [0,1] to (0,1). I recommend a "Minor Review"

Reviewer #4: 24) Line 74: it is rather unusual to write the abbreviation of the genus between brackets. Instead, the usual is to abbreviate after the first time it is mentioned.

25) Lines 81-4 and 153-5: reference is missing supporting claim. Additionally, the effects of climate change on dengue transmission is quite disputed and seems to strongly depend on the region and the methodology and criteria of analysis.

26) Lines 112-3: “support vector regression” is repeated in the list.

27) Line 123: “of one” sounds strange.

28) Line 127: should say “remains” instead of “remain”.

29) Lines 130-1: have LSTM been used elsewhere for dengue or other vector-borne disease? Mentioning it would provide a richer context.

30) There is a recurrent mentioning of “this is the first paper” or “this is the first time”. I value highlighting the novelties of the work, but this does not need to be done twice in the same paragraph (for instance, lines 135-46). Also, novelty should be highlighted in the context of the advantages it brings, not solely on being the first.

31) Lines 157, 159, and elsewhere: the standard is to leave a space between a magnitude and its unit.

32) Caption of Fig 1: “Vietnam” is unnecessarily repeated.

33) Be consistent in how regions are named. E.g., “Middle” or “Central”?

34) Line 186: “sunshine” refers to daylight or having no clouds? Please, explain in the text.

35) Line 185 and elsewhere: I prefer referring to “weather” instead of “climate” in this context, as climate refers to the average weather patterns along decades, while here you are addressing the particular meteorological conditions of each month.

36) Lines 213-4: how was overfitting assessed? Please explain in text.

37) Reference to figures: the text explains (and repeats) what the captions say. This conspires against clarity and conciseness. All references of the sort “Fig X shows...” or “In Fig X...” should be avoided. Instead, just mention, e.g., “LSTM showed better performance (Fig X)”. Try to avoid referencing figures in the discussion.

38) Lines 431-9: this paragraph sounds like part of Introduction or a justification in Methods, rather than Discussion.

39) Line 436: requiring “specific domain knowledge” is not a disadvantage of traditional models and, in fact, any analysis requires specific knowledge (for instance, machine learning methods).

40) Lines 452-5 and 472-474: this should be moved to Results; it is not properly discussion.

41) Lines 461-9: this paragraph seems out of the scope of this paper as it addresses data/results not shown and it delves into matters not within the objectives and analyses proposed. As is, this paragraph should be removed.

42) “Long term”: what is considered long term? What criteria are used for defining long term?

43) Line 480: syntax seems confusing or incorrect.

44) Conclusion paragraph: it looks like a summary, rather than a conclusion (except for lines 534-8). A further emphasis on the implications of the paper, beyond what has already been said in the rest of the discussion, could turn this into a proper conclusion and enrich the work.

45) Line 550: “a the” seems an error.

46) Reference formatting is not homogenous, nor it complies always with journal guidelines.

Reviewer #5: (No Response)

Reviewer #6: The manuscript is well written overall, some editorial and minor modifications to enhance clarity are the following:

Introduction

1. After lines 80 and 81, define reported cases whether cases suspected or confirmed are included.

2. Lines 110 and 113, it seems that “support vector regression” is repeated.

3. Lines 127 and 128, the authors state that the northern and central region of Vietnam “have not been touched”, this implies that no DF prediction models were assessed in both areas, however, have you considered the study by Colon-Gonzales F., et al. 2021 PLos Med study?, this study has evaluated predicting modes for DF in Vietnam and tested their model in the entire country.

4. Lines 138, 139, please add “out of 69” after “20”, this gives a better scope of the study.

5. Line 141, please provide if there is any data on which of the 12 climate factors used as predictors in your modeling, were the most frequently associated with better prediction outcomes. 

6. Line 142, consider adding the actual time frame for short and long-term DF incidence, 1 month and 3 months?

Materials and Methods

7. Line 164, a definition of “incident cases” is needed to clarify it the case counts included suspected cases or confirmed DF cases.

8. Lines 171, 172, the authors indicate that incidence of DF and death rates increased with temperature, and in June-October the rates were higher than in other months. Data supporting this observation is not provided in the manuscript, and whether this finding had any impact on their model development is not indicated anywhere either, “increased temperatures” should have been one of the best predictors from the 12 climate factors evaluated.

9. Lines 185-187, authors indicate that 12 meteorological factors were included in the study, but a clear description of the five types of measures would improve clarity, e.g. temperatures were measured as minimal, or maximal temperatures, where those “air temperature measured 2 meter above the grown, etc.?

10. Line 202, the “datasets had missing few data points”, in which provinces? Please indicate which provinces, have these missing points impacted negatively on the data even despite of the normalization the authors performed?

11. Line 206, please expand on the “different features and different value ranges” to improve clarity.

12. Line 212, 213, you referenced how the 12 meteorological factors were ranked per province, to define 2 top ranked factors. Accordingly, the supplemental S1 Table, the top 2 factors selected/province are listed, but a metric of the rank values are not provided. This information will add clarity to the results obtained.

13. Line 235, please add “in Malaysia” after “Kuala Lumpur” for clarity.

Results

14. Lines 298-300, only 6 provinces were selected to illustrate the prediction accuracy data, please explain why those provinces were selected and why the other 14 provinces were not. Additional information is needed to clarify whether the other 14 provinces have similar outcomes or perhaps opposite outcomes, this information will improve clarity.

15. Lines 300, 301, prediction lines for LSTM an LSTM-ATT are considered to have “very good prediction accuracies”, again, statistical data is needed to understand why it is considered a “very good prediction accuracy”. Although, authors indicate that the MAE and RMSE values (Figure 7) provide the metrics of fig 6, the fitted predicted incidence lines should be compared with the real incidence lines to see whether the predicted lines fall closer or under the 95% credible interval of the real incidence line. 

Discussion

16. Lines 432- 436, there are 4 limitations of traditional machine learning (ensemble and statistical) models that the authors discuss but no reference is provided. Please add references for these 4 limitations.

17. Lines 489-490, perhaps the study conducted by Colon-Gonzales F., et al. (2021 PLos Med), should be considered as they tested short (1 month) and long term (3 and 6 months) prediction models in the 63 provinces of Vietnam from over 19 years, from 2002-2020.

18. Lines 516-518, since missing cases and climate data are considered potential confounding factors, how do the authors address these factors in this study? Similarly, differences of performance have been found in some provinces compared with the rest, but a discussion on how the authors overcome this limitation is not provided.

Conclusion

19. Lines 530-531, the authors indicate that the three regions studied in Vietnam have significant differences in geographical and climate conditions, however, there is lack of information on these “geographical and climate conditions” for these three regions through the manuscript. Authors should consider providing this information early when describing the study in material and methods section. 

References

20. Under reference #3. Change “Organisation” for “Organization”

Reviewer #7: I have no recommendations to the authors or Editor. The paper is well written.

**Summary and General Comments**

Reviewer #1: The analysis is quite extensive and valuable, but revision is needed. Some figure can be removed, and some supplementary figure can be added for integrity of data. If the principle can be explained more clearly, like the choosing of climate factors, the article will be more persuading. Finally, the discussion which couldn’t address the objectives is unnecessary.

Reviewer #2: (No Response)

Reviewer #3: The manuscript is original. It has great methodological strength and statistical analysis processes. The application of methodological processes in public health could have a great impact, but I consider including environmental factors of housing and the presence of breeding sites, which could strengthen the models.

Reviewer #4: 47) Dengue literature is vast. However, I was surprised not finding any reference to the important and solid prediction work performed in Thailand, for instance. Check out:

- Reich et al. (2016) Challenges in real-time prediction of infectious disease- a case study of dengue in Thailand

- Lauer et al. (2018) Prospective forecasts of annual DHF incidence in Thailand, 2010-2014

- Kiang et al. (2021) Incorporating human mobility data improves forecasts of dengue fever in Thailand

And the multi-model challenge of forecasting dengue in Puerto Rico:

- Johansson et al. (2019) An open challenge to advance probabilistic forecasting for dengue epidemics.

48) Writing should be improved for clarity and correctness. Several points are mentioned above.

49) Content is mixed between sections, making reading more difficult. Several instances are mentioned above.

50) Some analyses should be enriched or extended. See points 6 and 18 above.

51) The subject and approach are interesting and relevant. Improving analyses as pointed above, improving presentation, and enriching the discussion would translate in a solid, clear, and relevant paper.

Reviewer #5: In this manuscript, the authors explored the potential of using deep learning methods to predict dengue monthly incidence. They compared the performance of multiple deep learning models, including convolutional neural network (CNN), Transformer, long short-term memory (LSTM), and attention mechanism-enhanced LSTM (LSTM-ATT) models. They used the Root Mean Square Error (RMSE) and Mean Absolute Error (MAE) indexes to compare these models.

I have below comments and suggestions:

(1) In page 5, the authors mentioned many other methods that are different from deep learning models. If I understand correctly, the basic idea of deep learning models is to perform complex regression to capture the complex dependency between data and predictors. As the authors have mentioned other types of simpler regression models, such as Poisson regression models, generalized additive models, I’d like to suggest the authors to compare the performance of using complex deep learning models versus that of using simpler regression models. This comparison will let us know whether using complex deep learning models can perform better than using simpler regression models.

(2) For predictors, the authors mainly considered climate factors. However, population immunity is an important predictor for dengue severe disease. For example, primary infection is known as the best predictor of developing severe dengue. The authors may wish to include population immunity as an important predictor.

(3) Is it possible to apply your method to dengue data in year 2020 and 2021?

Reviewer #6: The authors have compared current deep learning models to forecast Dengue Fever in three regions of Vietnam- CNNs, Transformer, and long short-term memory (LSTM); and they have successfully developed an attention mechanism- enhanced LSTM model (LSTM-ATT) that outperformed the previous modeling methods. I have mixed feeling about this paper. On one hand, the work is relatively impressive, with extensive datasets spanning 19 years, from 1997 to 2016, and top-notch competing deep learning methods were simultaneously tested. This study is bringing novel information on DF forecasting deep learning methods, and the results found can significantly improve current capacity to appropriately respond to predicted DF outbreaks within 1 and 3 months of actual outbreaks. However, there are some aspects that the authors may need to address to improve the overall quality of the manuscript. 

My first concern is about the apparent lack of information in the materials and methods section. The study area includes 20 provinces of three main regions in Vietnam, however geographical description of the three main regions studied are not provided; additional detailed description of the 12 meteorological factors used as predictors is needed, a first description of the prediction accuracy metrics “specificity, sensitivity, true positive, etc.” is needed to improve further data comparison and interpretation. 

Reading the result section, I noticed some relevant information seems to be lacking, which may improve the interpretation and significance of the work. Fig 1 depicts predicted incidences vs. real incidence as part of the 1-step forecasting accuracy, there is need of statistical values, e.g. 95% credible intervals, or IC values for the real incidence line depicted on the graphs. The Table S1 provide a list of the top 2 geometrical factors that ranked best among 12 factors evaluated per each province. Actual rank values could be added for each selected factor for data completeness.

Additionally, not all relevant competing papers were discussed- especially one that tested a superensemble forecasted method in the entire country of Vietnam, this paper was published very recently (Colon-Gonzales F., et al. March 2021) and its findings could inform the current study. So, although the present study could be the first-to adapt deep learning methods to forecast DF based on climate factors, other work has been done with similar outcomes and in a more complete dataset including the 63 provinces of Vietnam.

Information on the authors' Institutional Review Board or an equivalent committee approval for research involving human participants may be needed. A brief overview of the ethics reporting should be provided in the Methods.

Reviewer #7: The study is original and revealed the usefulness of deep learning models for climate-based Dengue Fever forecasting, in predicting epidemics before three months with the employment of environmental data. It is very well written and the authors included limitations of the methods in the discussion section.

PLOS authors have the option to publish the peer review history of their article (what does this mean?). If published, this will include your full peer review and any attached files.

Reviewer #1: Yes: Liyun Jiang

Reviewer #2: Yes: Sumaira Zafar

Reviewer #3: No

Reviewer #4: No

Reviewer #5: No

Reviewer #6: No

Reviewer #7: No
---

## [Decision Letter · Decision Letter 1]

6 Feb 2022

Dear Associate Prof. Tran Thi,

Thank you very much for submitting your manuscript "Deep learning models for forecasting dengue fever based on climate data in Vietnam" for consideration at PLOS Neglected Tropical Diseases. As with all papers reviewed by the journal, your manuscript was reviewed by members of the editorial board and by several independent reviewers. The reviewers appreciated the attention to an important topic. Based on the reviews, we are likely to accept this manuscript for publication, providing that you modify the manuscript according to the review recommendations. 

- The language needs to be checked carefully.

- Please upload the analyzed data as a supplementary file for easy reproducibility of the reported findings. This will be a criteria for acceptance. 

- Please write the section name and lines/pages number for every change you make.

Sincerely,

Samuel V. Scarpino

Deputy Editor

Samuel Scarpino

Deputy Editor

- The language needs to be checked carefully.

- Please upload the analyzed data as a supplementary file for easy reproducibility of the reported findings. This will be a criteria for acceptance. 

- Please write the section name and lines/pages number for every change you make.

Reviewer's Responses to Questions

**Key Review Criteria Required for Acceptance?**

**Methods**

-Are the objectives of the study clearly articulated with a clear testable hypothesis stated?

-Is the study design appropriate to address the stated objectives?

-Is the population clearly described and appropriate for the hypothesis being tested?

-Is the sample size sufficient to ensure adequate power to address the hypothesis being tested?

-Were correct statistical analysis used to support conclusions?

-Are there concerns about ethical or regulatory requirements being met?

Reviewer #2: Yes

Reviewer #4: (No Response)

Reviewer #5: (No Response)

Reviewer #6: Yes, objectives and study design are clear and appropriate. There are no concerns about ethical or regulatory requirements.

Reviewer #7: The methods are clearly described and testing in municipal scale instead of the entire country was very important to find a model of deep learning methodologies to associate environmental parameters with dengue fever occurrence.

**Results**

-Does the analysis presented match the analysis plan?

-Are the results clearly and completely presented?

-Are the figures (Tables, Images) of sufficient quality for clarity?

Reviewer #2: Authors fairly incorporated all the comments from reviewers

Reviewer #4: (No Response)

Reviewer #5: (No Response)

Reviewer #6: Results are now more clear and complete. Authors have included suggested modifications.

Reviewer #7: The results are very well presented with sufficient tables and images.

**Conclusions**

-Are the conclusions supported by the data presented?

-Are the limitations of analysis clearly described?

-Do the authors discuss how these data can be helpful to advance our understanding of the topic under study?

-Is public health relevance addressed?

Reviewer #2: Authors fairly incorporated all the comments from reviewers

Reviewer #4: (No Response)

Reviewer #5: (No Response)

Reviewer #6: Conclusions are supported by the data, suggested modifications to methods, results, and discussion session now support the conclusions.

Reviewer #7: Conclusions were addressed to public health relevance and is supported by the presented data.

**Editorial and Data Presentation Modifications?**

Reviewer #2: Accept

Reviewer #4: (No Response)

Reviewer #5: (No Response)

Reviewer #6: None

Reviewer #7: No recommendations

**Summary and General Comments**

Reviewer #2: (No Response)

Reviewer #4: The points in my review were adequately addressed. I thank the authors for the effort and clarity in responding the numerous points.

Reviewer #5: (No Response)

Reviewer #6: The authors have appropriately addressed most of the observations and have incorporated modifications suggested throughout the manuscript. This manuscript if accepted for publication can be a relevant contribution to the community, the forecasting models tested in this work have the potential to enhance public health responses to in the control of climate-impacted diseases.

Reviewer #7: The presented work is original and enable prediction of dengue fever outbreaks with a time of three months. The regional scale used is original and enabled the important findings of the work.

PLOS authors have the option to publish the peer review history of their article (what does this mean?). If published, this will include your full peer review and any attached files.

Reviewer #2: No

Reviewer #4: No

Reviewer #5: Yes: Lin Wang

Reviewer #6: No

Reviewer #7: No

Figure Files:

Data Requirements:

Reproducibility:

References

---

## [Decision Letter · Decision Letter 2]

17 May 2022

Dear Associate Prof. Tran Thi,

We are pleased to inform you that your manuscript 'Deep learning models for forecasting dengue fever based on climate data in Vietnam' has been provisionally accepted for publication in PLOS Neglected Tropical Diseases.

Best regards,

Mohamed Gomaa Kamel

Associate Editor

Samuel Scarpino

Deputy Editor

Reviewer's Responses to Questions

**Key Review Criteria Required for Acceptance?**

**Methods**

-Are the objectives of the study clearly articulated with a clear testable hypothesis stated?

-Is the study design appropriate to address the stated objectives?

-Is the population clearly described and appropriate for the hypothesis being tested?

-Is the sample size sufficient to ensure adequate power to address the hypothesis being tested?

-Were correct statistical analysis used to support conclusions?

-Are there concerns about ethical or regulatory requirements being met?

Reviewer #5: (No Response)

Reviewer #6: Objectives and study are clearly articulated, and study design appropriately addressed the objectives.

Reviewer #7: (No Response)

**Results**

-Does the analysis presented match the analysis plan?

-Are the results clearly and completely presented?

-Are the figures (Tables, Images) of sufficient quality for clarity?

Reviewer #5: (No Response)

Reviewer #6: Results are now clearly and completely presented. Figures and tables are of sufficient clarity. Authors fairly incorporated recommendations and suggestions made.

Reviewer #7: (No Response)

**Conclusions**

-Are the conclusions supported by the data presented?

-Are the limitations of analysis clearly described?

-Do the authors discuss how these data can be helpful to advance our understanding of the topic under study?

-Is public health relevance addressed?

Reviewer #5: (No Response)

Reviewer #6: Conclusions are supported by the findings of the study. Limitations are now clearly stated and discussed. Public health relevance is addressed.

Reviewer #7: (No Response)

**Editorial and Data Presentation Modifications?**

Reviewer #5: (No Response)

Reviewer #6: Accept

Reviewer #7: (No Response)

**Summary and General Comments**

Reviewer #5: (No Response)

Reviewer #6: The authors have successfully incorporated suggested modifications which enhanced the clarity of the manuscript. This manuscript provides insightful contributions to the community, the forecasting models analyzed in this study can improve surveillance and control of climate-sensitive diseases.

Reviewer #7: (No Response)

PLOS authors have the option to publish the peer review history of their article (what does this mean?). If published, this will include your full peer review and any attached files.

Reviewer #5: No

Reviewer #6: **Yes: **Neida Mita-Mendoza

Reviewer #7: No

---

## [Editor Report · Acceptance letter]

7 Jun 2022

Dear Associate Prof. Tran Thi,

We are delighted to inform you that your manuscript, "Deep learning models for forecasting dengue fever based on climate data in Vietnam," has been formally accepted for publication in PLOS Neglected Tropical Diseases.

Best regards,

Shaden Kamhawi

co-Editor-in-Chief

Paul Brindley

co-Editor-in-Chief
